# Insights into the implementation of a whole genome sequencing report form (SRF) to reduce nosocomial SARS-CoV-2 in UK hospitals within an unfolding pandemic: A qualitative process evaluation using normalisation process theory

Ruth Leiser[1], Julie McLeod[2], Fiona Mapp[3], Oliver Stirrup[3], James Blackstone[4], Christopher J.R. Illingworth[5], Gaia Nebbia[6], James R. Price[7], Luke B. Snell[8], Tranprit Saluja[9], Judith Breuer[10], Paul Flowers[1]*

1 Psychological Sciences and Health, University of Strathclyde, United Kingdom, 2 School of Health and Life Sciences, Glasgow Caledonian University, Glasgow, Scotland, United Kingdom, 3 Institute for Global Health, UCL, London, United Kingdom, 4 Comprehensive Clinical Trials Unit, UCL, London, United Kingdom, 5 MRC-University of Glasgow Centre for Virus Research, Glasgow, United Kingdom, 6 Department of Infection, Guy's and St Thomas' Hospital NHS Trust, United Kingdom, 7 Department of Global Health and Infection, Brighton and Sussex Medical School, University of Sussex, Brighton, United Kingdom, 8 Kings College London and Department of Infectious Diseases, London, United Kingdom, 9 Sandwell & West Birmingham Hospitals N.H.S. Trust, Birmingham, United Kingdom, 10 Department of Infection, Immunity and Inflammation, UCL Great Ormond Street Institute of Child Health, United Kingdom

* Paul.flowers@strath.ac.uk

## Abstract

### Background

Here we report on a process evaluation conducted as part of a large multisite non-randomised trial of the effectiveness of a novel whole genome sequence report form (SRF) to reduce nosocomial SARS-CoV-2 through changing infection prevention and control (IPC) behaviours during the COVID –19 pandemic. We detail how the SRF was implemented across a heterogeneous purposive sub-sample of hospital trial sites (n=5/14).

### Methods

We conducted in-depth interviews from diverse professional staff (N=39). Deductive and inductive thematic analysis initially explored participants' accounts of implementing the SRF. The resulting themes, concerning the way the SRF was used within sites, were then coded in relation to the key tenets of normalisation process theory (NPT).

### Results

Factors that enabled the implementation of the SRF included: elements of the context such as health care professional passion; the existence of whole genome sequencing

**Data availability statement:** The data supporting the findings of this study are small and rich in qualitative detail. Due to the sensitive and confidential nature of the data, as well as explicit consent restrictions from several participants, the data cannot be shared with others beyond the original research team. For further inquiries, correspondence can be directed to Dr. Paul Flowers at Paul.Flowers@strath.ac.uk or Professor Ben Jones, Director of Research, at Benedict.Jones@strath.ac.uk.

**Funding:** This work was supported by funding from the Medical Research Council (MRC) part of UK Research & Innovation (UKRI), the National Institute of Health Research (NIHR) [grant code: MC_PC_19027], and Genome Research Limited, operating as the Wellcome Sanger Institute. The funders had no role in study design, data collection and analysis, decision to publish, or preparation of the manuscript.

**Competing interests:** The authors have declared that no competing interests exist.

**List of abbreviations:** CMO = Context-Mechanisms-Outcome; COG-UK = COVID-19 Genomics UK consortium; COG-UK HOCI = COG UK's Hospital-Onset COVID-19 Infections Study; HCAIs = Healthcare-associated infections; HOCI = Hospital-Onset COVID-19 Infection; SRF = Sequence report form; IPC = Infection prevention and control; SARS-CoV-2 = the virus that causes COVID-19; NPT = Normalisation process theory; WGS = Whole genome sequence

(WGS) infrastructure; effective communication channels, the creation of new connections across professionals and teams; the integration of SRF-led discussions within pre-existing meetings and the ability of a site to achieve a rapid turnaround time. In contrast, we found factors that constrained the use of the SRF included elements of the context such as the impact of the Alpha-variant overwhelming hospitals. In turn, dealing with COVID-19 breached the limited capacity of infection prevention and control (IPC) to respond to the SRF and ensure its routinisation.

## Conclusion

We show preliminary support for this SRF being an acceptable, useable and potentially scalable way of enhancing existing IPC activities for viral respiratory infections. However, the context of both the trial and the alpha wave of COVID-19 limit confidence in these insights.

## Clinical trial number

https://www.isrctn.com/ISRCTN50212645, Registration date 20/05/2020

## Contributions to the literature

### What is known about this issue already?

Very little is known about the use of sequence report forms (SRF) that translate genomic insights concerning the transmission of infectious disease in order to inform infection prevention and control (IPC) responses. Genomic data and SRFs are not routinely used in IPC responses to infections in UK hospitals (either viral, bacterial or fungal). In the context of the COVID-19 pandemic it was possible to evaluate the use of SRFs on IPC for the first time.

An earlier behavioural analysis of the SRF used within our multisite non-randomised trial highlighted that in relation to detecting and reducing transmission of SARS-CoV-2 the SRF was acceptable, that its content worked as anticipated, and that theoretically it should have led to changes in IPC response within the COVID pandemic

However, results from our multisite non-randomised trial suggested that the SRF only sometimes informed an IPC response and, overall, no effect was detected on the incidence of nosocomial transmission (the primary trial outcome)

Very little is known about what drives the effective implementation of SRFs within hospital contexts

### What does this paper add?

The paper shows, for the first time, the potential of an SRF to inform IPC responses within an unfolding viral pandemic

This paper highlights the profound impact of a surge in cases of virus infection, associated with the SARS-CoV-2 Alpha variant, upon IPC capacity in general and upon the implementation of the SRF in particular

The SRF drove the development of new inter-professional practice leading to IPC responses to the SRF only where and when local conditions permitted. This

interprofessional practice consisted of effective modes of communication, newly connected networks of individuals and teams, and timely meetings

Recommendations: Further transdisciplinary work on the use of SRFs for a range of infectious diseases is needed, as is more detailed research concerning their optimal design. Future research must also focus on the communications, and implementation packages, that must surround the use of SRFs.

## Background

During the COVID-19 pandemic, nosocomial transmission of SARS-CoV-2 presented a significant problem for patients and healthcare staff, as it negatively impacted hospital function [1,2,3]. Accurate identification of nosocomial infection was important in targeting overstretched infection prevention and control (IPC) resources to where they were needed most.

To complement ongoing epidemiological investigations into outbreaks, novel attempts were made to harness insights from the whole genome sequencing (WGS) of SARS-CoV-2 to assist with identifying and managing nosocomial infection [4,5,6,7]. When there is sufficient variation in pathogen genotypes, WGS can identify linked clusters of cases and, when combined with records of patient location and movement, can enable further focused IPC activity to prevent onwards transmission. For many viral infections at least, for such genomic insights to be useful and reduce ongoing transmission, it is vital that they are delivered in a timely fashion, through a fast 'turnaround time' [8]. Given that genomic outputs are often complex and require specialist/technical expertise/knowledge to decipher, it is also important that such insights are effectively translated so they can be easily understood by front line staff to enable timely action.

As part of the COVID-19 Genomics UK (COG-UK) consortium, the Hospital-Onset COVID-19 Infections Study (COG-UK HOCI) [9,10] sought to address these two issues (i.e., both timeliness and effective translation). This context, of using an SRF within an unfolding viral pandemic, is important to stress given how different using an SRF may prove to be for either bacterial or fungal infections. For viral infections, for example, turnaround time may be particularly important because incubation and transmission are expedited.

For the COG-UK HOCI study a simple sequencing reporting form (SRF) was developed to translate genomic insights into an easy-to-understand form and change IPC behaviour [11]. This was then trialled in a non-randomised prospective study to assess the relative impact of diverse turnaround times (i.e., rapid vs slow) of the translated genomic insights [12]. Overall, the results of the trial were mixed. The trial's primary analysis suggested no direct impact of the SRF on the incidence of nosocomial transmission. However, a *per protocol* sensitivity analyses suggested the SRF did inform IPC response to Hospital-Onset COVID-19 Infection (HOCI), particularly when returned within five days. Following this, the SRF was further analysed using tools from behavioural science [11] and there was broad support for it working in the ways it was intended to, suggesting no major problems with its content.

Given that the SRF's content was largely found to work as intended yet, when trialled, it had no direct impact on the incidence of nosocomial infection, there is a need for further detailed analysis of its implementation to understand how and why the SRF produced differential outcomes across trial sites. This study seeks to understand the implementation of the SRF through the lens of normalisation process theory (NPT) [13]. NPT has been specifically

developed to theorise implementation processes surrounding given interventions. NPT focuses on the many overlapping processes, decisions, beliefs, behaviours, relationships, and roles that enable intervention implementation. Extensive research with NPT has examined the implementation of a variety of healthcare interventions to date [14; 15]. Problems with implementation are typically attributed to a number of factors, including slow individual-level behaviour change, obstinate organisational-level norms, values and practices, and the role of dynamic ever-changing contexts [16]. Together these issues highlight the need to understand the implementation of our SRF in depth and detail.

## Aims

To systematically explore the implementation of the SRF and detail useful learning for future work evaluating sequencing reporting.

### Research questions

1. How did the context shape the ways in which the SRF was implemented?

2. Which implementation mechanisms were important for understanding how the SRF effected outcomes?

3. How did people using the SRF imagine its effect on trial outcomes?

## Methods

### Overview of design

This study took a qualitative approach to intervention evaluation using NPT, as part of a multi-level process evaluation [17]. Here we explore the perspectives of diverse staff recruited from five of fourteen sites involved in the wider COG-UK HOCI study. Interviews concerned the use of an SRF designed to reduce nosocomial COVID-19 infections by changing the location and intensity of infection prevention and control (IPC) behaviour. Data was analysed using inductive and deductive thematic analysis. Themes were subsequently explored in relation to key tenets from NPT.

### Study details

Full details of the HOCI trial and development of the SRF can be found elsewhere [9,10]. Fourteen sites were enrolled in the trial, from October 2020 to April 2021 (during the UK's Alpha wave). The SRF was implemented in each site over a number of weeks and three distinct phases. First, sites recorded baseline case rates and IPC practice over a period of four weeks (phase 1); subsequently, sites employed genome sequencing for hospital onset COVID-19 cases, with a 'rapid' turnaround aim of 48hrs (phase 2) and a 5–10-day turnaround period (phase 3), respectively. WGS output, when coupled with epidemiological meta-data, comprised the SRF intervention. Following the implementation of the SRF, qualitative data collection began.

### Participants

Of the 14 sites in which the SRF was trialled, five were selected for qualitative data collection. Site selection was informed by variation in numerous aspects, including hospital size and case rates. Within each site, participants were selected purposively to ensure diversity across job role and familiarity with WGS. Data was not collected about who refused to take part. Sample

size ranged from six to nine participants from each site, with the overall sample comprising 39 staff. Participant roles were varied (e.g., WGS lab manager, clinical fellow, research nurse).

## Ethics statement and data collection

Ethical approval was given by Cambridge South Research Ethics Committee (20/EE/0118) who reviewed the study documentation (participant information sheet, consent form, topic guide and standard operating procedure for data collection). All methods were carried out in accordance with relevant guidelines and regulations and informed consent was obtained from all research participants.

Potential participants were provided with an information sheet and consent form, before one-to-one interviews were arranged. Verbal consent was attained during the recording prior to data collection beginning by FM asking the participant if they agreed to a series of statements relating to how data were being collected and used for the study. The date and time consent was obtained were recorded and witnessed by FM and the participant. Consent for the data to be included in the study was reconfirmed at the end of the interview and an electronic copy of the completed consent form was sent to participants and stored securely by UCL.

Semi-structured interviews were conducted virtually, lasting between 30 and 90 minutes, all by one of the authors with considerable interview expertise (FM). Participants were largely unknown to the interviewer but they knew why the research was being conducted and that the interviewer had an unbiased interest in learning about implementation. A topic guide was employed to guide the interview – questions were intended to explore participants' experience of the trial and their perspectives on using the SRF. The topic guide was used

**Table 1. An overview of the central tenets of Normalisation Process Theory.**

| CMO domain | NPT construct | Explanation |
|---|---|---|
| **Contexts** Labile features of the environment that effect implementation | Strategic intentions | The ways in which contexts shape the development and planning of an intervention |
| | Adaptive execution | The ways in which users of an intervention can engineer work-arounds to make it function in practice |
| | Negotiating capacity | The extent to which users of an intervention can make it fit into their usual practice |
| | Reframing organisational logics | The ways in which "social structural and social cognitive resources" influence and impact the implementation environment |
| **Mechanisms** The collaborative work needed to implement an intervention | Coherence | The ways in which users of an intervention understand it and its potential benefits |
| | Cognitive participation | The extent to which users of an intervention believe in its potential and are willing to commit to implementing it |
| | Collective action | The ways in which users of an intervention work to facilitate its successful function |
| | Reflexive monitoring | The ways in which users of an intervention appraise its outcomes/success |
| **Outcomes** The practical effects of implementing the intervention | Intervention Performance | The practices that have changed as a result of the intervention over time and across settings |
| | Relational Restructuring | The ways relationships are changed by implementing an intervention |
| | Normative Restructuring | The ways in which implementation changes norms, rules and resources |
| | Sustainment | The extent to which interventions become incorporated into practice |

flexibly in the interviews. Topics covered a range of issues such as how the particular site was responding to COVID at the time of interview, the participants' role in relation to using the rapid sequencing report, challenges and opportunities associated with the trial, perspectives on the specific nature of the problem the SRF was designed to address, experiences of starting to use the SRF, thoughts on the detail of the SRF, perspectives on how the SRF may have hindered COVID management in the hospital, accounts of how the SRF affected IPC, patient pathways, hospital acquired COVID, accounts of work/life balance, stress and anxiety. Interviews were audio recorded and transcribed by an external company, then transcripts were anonymised.

## Analysis

Initial analysis was an iterative process involving both deductive and inductive thematic analysis [18]. First, authors PF and FM engaged in multiple data readings and discussions. After this, an initial coding frame was developed containing nine broad categories of data. (1. the perceived problem addressed by the SRF (three sub-codes); 2. details of the context (eleven sub-codes); 3. the acceptability of sequencing for managing infectious disease per se (seven sub-codes); 4. the acceptability of sequencing for managing COVID-19 (seven sub-codes); 5. acceptability of the SRF (seven sub-codes), 6. perspectives of implementing the SRF (seven sub-codes); 7. perspectives on the outcomes and effects of using the SRF (eleven sub-codes); 8. the personal impact of COVID-19 (five sub-codes); and 9. material relating to the trial (seven sub-codes). Some of these nine broad categories were pre-specified (e.g., an *a priori* deductive focus on the trial for example) others were identified from the data and initial discussions (i.e., the acceptability of sequencing).

In separate codebooks, one for each of the five sites, a wider team of five qualitative researchers then coded data from the transcripts to relevant sections of the coding-frame. All data from each of the separate codebooks were then collated into one codebook with data from all five sites and audited by PF and FM.

Following this initial categorisation of the data, using data from sections five, six, and seven of the coding- frame, inductive themes relating to implementation of the SRF were generated by PF and audited by RL. These included new data-driven themes such as '*Passion, will, and infrastructure coalesced around the SRF*'. RL and PF then worked collaboratively to map these data-driven themes to NPT constructs and domains (see Table 1) through iterative discussion. Where disagreement occurred, extensive discussion took place to reach a consensus. These inductive themes and their mapping to NPT were then discussed with the wider interdisciplinary trial team.

## Theoretical frameworks to understand intervention implementation

NPT provides a framework for understanding the implementation of diverse interventions [13]. It has been used to explore the early implementation of interventions [19] often within process evaluations [20,21] and, more rarely, focussing on the longer-term maintenance of interventions [22]. NPT often follows the configuration of Context-Mechanisms-Outcome (CMO), popularised by Realist evaluations [23]. This configuration emphasises the importance of viewing an intervention not as something that either dichotomously 'works' or 'does not work', but as something that is *enacted* by people within particular contexts and that, through a range of mechanisms, ultimately changes outcomes to varying degrees.

*Context* refers to aspects of the social, temporal and spatial circumstances into which interventions (here the SRF) are introduced. Context can be particularly complex within process evaluations because effective interventions that become embedded over time are often

Table 2. An overview of the data driven themes and how they relate to normalisation process theory.

| Implementation Theme | Summary of data driven findings | NPT *constructs* | Implications for the future implementation of SRFs within and outwith future pandemics |
|---|---|---|---|
| **Context** | | | |
| **Passion, will, and infrastructure coalesced around the SRF** | The wider context of COVID-19 facilitated SRF implementation.<br>It did this by coalescing longstanding interest in WGS and capitalising on available infrastructure and resources enabled by the COVID-19 pandemic (e.g., wider genomic surveillance).<br>There was a notable interest in how the SRF could be important in the future as well as with the present. | Reframing organisational logics<br>Strategic intentions | Identify and capitalise on individuals across various organisations with an interest in exploring and demonstrating the applied value of WGS<br>Maintain a national and regional sequencing infrastructure for outbreak management and pandemic preparedness<br>Explore the need for maintaining hospital and community sequencing infrastructure to enable applied WGS |
| **The impact of the Alpha variant**: 'Surrounded and deluged' | The context of the COVID-19 Alpha variant severely hindered implementation of the SRF – an unmanageable set of circumstances worked against the routinisation of SRF implementation | Reframing organisational logics<br>Strategic intentions | Depending on the nature and scale of outbreaks, or future pandemics, consider if there are actually realistic opportunities to implement new ways of working or whether such attempts should be made at another time |
| **Breaching capacity to act**: 'Everything grinds to a halt' | The pandemic context hindered implementation of the SRF as healthcare staff were overburdened, resources were stretched to capacity, and the ability to adapt and engineer workarounds to embed the SRF were often severely diminished | Adaptive execution<br>Negotiating capacity | Depending on the nature and scale of future outbreaks or pandemics, consider if it is possible to generate new resource to offset diminishing capacity to respond and to enable the implementation of innovations |
| **Mechanisms** | | | |
| **Effective communication catalyses a chain of action** | Success of the SRF depended on the extent to which teams communicated effectively and executed a collective response to implementing the SRF<br>Effective communication about the SRF could catalyse a chain of action across diverse teams | Collective action | Ensure an extensive range of staff within hospitals are aware of the SRF and understand how it works (e.g., including issues of its time criticality)<br>Encourage hospitals to generate locally relevant plans (e.g., effective data sharing and timely meetings) that can ensure that SRF insights are rapidly translated into IPC action |
| **New connections catalyse a chain of action** | The relative success of the SRF relied upon the forging, and subsequent maintenance, of new relationships between previously disparate staff members and teams | Collective action | Prepare and ready the full range of professionals that the SRF's insights will affect and find ways of engaging them as a group.<br>Ensure the full range of staff affected by the SRF know and understand what the SRF can do and understand that its potential effectiveness is time critical. |
| **Meetings matter and establish a chain of action** | The relative success of the SRF depended on the extent to which sites implemented, and then routinised, meetings which formalised the process of people working together to use insights from the SRF | Collective action | Consider the best combination of focussed interactions that can facilitate timely action across the chain of professionals involved in using the SRF to change IPC.<br>Consider using existing meetings and associated communication structures to routinise or standardise SRF-focussed action. |
| **Time criticality and acknowledged value of the SRF** | Relative success of the SRF relied upon staff understanding its value and the unique contribution it could make beyond usual practice.<br>This understanding, however, also extended to an awareness of its time-dependent diminishing returns. In other words, some staff who understood how time critical the SRF was subsequently felt confident to ignore its insights when there was a long turnaround time. | Coherence<br>Cognitive participation | Ensure that all communication and potentially intervention branding and messaging all highlights time criticality of the SRF and the translation of its insights (e.g., highlight 'accelerated' or 'expediated')<br>Ensure that communications, and meetings have a distinct focus on reducing turnaround time, and associated time to IPC action<br>Consider developing a clear case study for teams to understand the importance of the rapid feedback loop that can drive effective improvements to IPC |
| **Understanding the importance of SRF attribution** | Success of the SRF relied upon a complex multi-layered network of staff working together to enact it. However, this 'layering' rendered the effects of SRF invisible to some. In other words, some staff may never have been aware their actions were driven by the SRF | Coherence | Ensure that a full range of staff including front line IPC staff, cleaners, and administrators are aware of and understand what the SRF is doing and what it is hoping to achieve.<br>Ensure there is a shared understanding of what the SRF is uniquely delivering |
| **Outcomes** | | | |
| **Partial success** | Participants talked of their perception of the partial success of the SRF. | *Premature given time limited trial* | Consider formalising the connections and relationships that are needed to constitute the chain of actions that stems from the SRF through standardised responses to the SRF |

designed specifically to change the very context into which they are implemented. *Mechanisms* are the means by which interventions have an impact. Pawson & Tilley [23] stress the importance of how people understand an intervention – and how it relates to them – and the ways they work to enact it. In this way mechanisms are psychosocial and sociocultural relating to the conduct of both individuals and teams. *Outcomes* relate to the effects of implementing an intervention. Outcomes here are not the same as trial outcomes (e.g., primary outcome) but relate to implementation. They relate to how the mechanisms may eventually lead to changes in practices, relationships, social norms and the durability of these Changes over time.

Table 1 shows how under the three domains of *Context*, *Mechanism*, and *Outcome,* the NPT approach spans 12 further constructs each focussing on well-established important aspects of intervention implementation. Within the *Context domain*, NPT details four relevant constructs: Strategic Intentions, Adaptive Execution, Negotiating Capacity, and Reframing Organisational Logics. The *Mechanism* domain also consists of the four constructs: Coherence, Cognitive Participation, Collective Action, and Reflexive Monitoring. Finally, within the *Outcomes domain*, four further constructs exist: Intervention Performance, Relational Restructuring, Normative Restructuring, and Sustainment (Normalisation).

## Results

Our findings (see Table 2) address our research questions and are structured within the Context-Mechanism-Outcome framework.

### RQ1: How did the context shape the ways in which the SRF was implemented?

**Passion, will, and infrastructure coalesced around the SRF: 'Genomics has come of age'.** The context enabled the implementation of the SRF through the alignment of two key factors. Firstly, a pre-existing appetite to explore how sequencing could be used to manage healthcare-associated infections (HCAIs), and secondly, the availability of considerable infrastructure arising from the commencement of the COG-UK programme in 2020 to monitor emerging variants of SARS-CoV-2 across both communities and hospitals [24]. Although this alignment emerged during COVID-19, data often also spoke directly to the future ('*building a legacy,*' Site 1; '*a foundation for other viruses,*' Site 4). The extract below shows these elements and signals the considerable struggle of early implementation: -

> *In a time of need, it might be the right time to really push down these barriers [to using sequencing for IPC] because actually, you probably lose the will to live if it wasn't an emergency, I think at least we're sort of pushing them down for the future, if not for now. (Site 1)*

Passion and will to examine if this SRF would work in the COVID pandemic context was also reflected in several participants being willing to go above and beyond what would normally be expected of them professionally in order to get the SRF up and running.

> *It made me feel I was useful [in the pandemic response]. And that I could… I wasn't just not, I wasn't just sitting around, I was able to provide something. (Site 3)*

> *I'm spending time which I don't have on it and at weekends and things like that…… I think my partner hasn't seen a lot of me (Site 1)*

There is a strong resonance between this theme and NPT's contextual domain of "Reframing organisational logics" wherein 'existing social structural and social cognitive resources shape the implementation environment'. It also chimes with NPT's contextual domain of "Strategic intentions". Here, both data and theory draw attention to these facilitative aspects of the wider context of the COVID-19 pandemic; it enabled the implementation of the SRF at structural, organisational and intrapersonal levels. It is likely that outwith the COVID-19 pandemic, a trial of an SRF with a focus on the timeliness of the turnaround time would not have been possible.

**The impact of the Alpha variant: 'Surrounded and deluged'.** This theme shows how the temporal and epidemiological context was fundamentally challenging to the implementation of the SRF. Elements of this context included an unpredictable and increasingly complex work environment (working '*90 miles an hour*' Site 3); high levels of staff illness; staff burnout (there was '*a lot of trauma*' Site 3); the sheer volume of patients with COVID-19 ('*we shot off didn't we, we got stratospheric*' Site 4). Participants talked of this context as a '*vicious circle*' (Site 1) and that they were in '*dire straits*' (Site 3): -

> *Well, we're in a bad situation with Covid-19 in [city]. We've had many hospital onset infections across the whole of [NHS Trust], and that has affected both the patients in wards allround the hospital... and it's also obviously affected our staff as well, many of whom have been infected and also had to take time off for family members being infected as well. In terms of the sort of volume of… I mean… it's everywhere and very difficult to control. And actually we've been swamped with it […..] just the sheer volume and the numbers.(Site 1, 628)*

There is little immediate resonance between this theme and the theoretical content of NPT. However, it is possible to consider it as representing the unravelling of "Organisational logics" and the diffusion of "Strategic intentions". Typical hospital function was severely challenged by the scale of disease and death. As a result, the capacity to implement the SRF was therefore deeply compromised.

**Breaching capacity to act: 'Everything grinds to a halt'.** This critically important theme relates to a sense of the imposition of a COVID-related cap on hospital function: there were limits to being capable of responding to the changing COVID-19 situation, and relatedly, limits to the possibility of responding to SRF-delivered insights.

> *The infection control team, because the fact that there's been so much, so many cases, obviously they've been completely overworked, and so have probably been less able to make reactive decisions based on sequencing results, as they've just not got the capacity to deal with all the outbreak areas that we've had (Site 2)*

This theme resonates with two contextual domains within NPT: "Adaptive execution" and "Negotiating capacity". "Adaptive execution" has particular resonance with this theme of *breaching capacity to act*. This NPT construct focuses on how context shapes actors' ability to adapt the immediate environment to execute an intervention as intended. Participants talked at length of their diminished capacity to act: resources were stretched thin, staff were overburdened, '*decimated with Covid related stuff unfortunately*' (Site 3), and wriggle room to initiate or reinforce the use of the SRF was either labile or limited.

> *The stressors on the organisation are such that I don't think it's a, I don't think that the knowledge that transmission has occurred is leading to any intervention that would prevent further transmissions occurring, I think partly because pretty much everything that we've got the resource to do is being done….I don't think the rapid turnaround is proving critical at the moment because*

*of the weight of numbers, for the reasons that I just said, I don't think we've got the resource to respond to it, and I think we're doing pretty much all we can anyway. (Site 2, 694)*

## RQ2. Which implementation mechanisms were important for understanding how the SRF effected outcomes?

**Effective communication catalyses a chain of action: 'We've all come together to discuss those things and to look at the reports'.** Communications, and the subsequent chain of actions flowing from them, constitute an important theme. On a fundamental level, the SRF was only a report - yet it had to, and did at times, catalyse people and a series of processes to enable changes to IPC. In this way, for the SRF to work, there had to be an alignment of diverse people (e.g., the bioinformatician, the microbiologist, the IPC team) and a constellation of congruent sequential actions: -

*When we were in a flow of, 'okay, we've got this patient come through', this is the report, and speaking to [HOCI site PI], and then the nurses reacting to that, and making decisions based off of what we found in the reports, it was really interesting just to see the link between all of it, and just, how it can help (Site 5)*

Some sites effectively capitalised on the coterminous turn to remote working and found ways of using digital platforms, such as Teams, as a focal point – or hub – '*focusing in on what the actual team need to do their job*' (Site 1) to ensure a collective response to the SRF. Across the five sites and across time, the degree to which SRF-related communication catalysed IPC action differed. Not all sites managed to establish the 'flow' detailed above. Equally, over time, certain communication routes sedimented in response to the changing parameters of what was actually possible to do given the situation on the ground: -

*When we first started the rapid phase I was then sending every single report to all the infection control nurses with interpretation, because we had so many, they asked me to stop because … I was basically clogging up their inboxes, [.....] so I've bowed to their request, I've stopped sending those requests to them, I only send them where I think it's [the SRF] actually going to make a tangible difference. – (Site 2, 694)*

This theme clearly relates to the NPT constructs of "Collective action" – stressing the work that teams must do to make an intervention work in the way it is intended. However, there was a sense that this mechanism was particularly fragile and dependent on what local circumstances could permit through the progression of the Alpha variant wave.

**New connections catalyse a chain of action.** This theme highlights that success in the early implementation of the SRF initially relied on forging, and then sustaining, connections across teams within the hospital. New professional relationships and novel ways of working emerged – for example, between bioinformaticians, microbiologists and IPC nurses. The extract below captures this sense of connections being made '*I think it's [the SRF] enabled the development of collaborations that didn't previously exist, so yeah, I mean it certainly has positives there that are long-term' (Site 2, 694)*. New connections were not simply established and then remained static; they were dynamic and processual, as diverse staff understood the SRF and what it did, there was increased buy-in across different teams:

> *They [IPC nurses] definitely became interested and especially as it became clearer and clearer that data we were generating was useful and timely, yeah, they definitely became very keen and involved, it became more of a two-way process (Site 3)*

In some sites, these new connections began to routinise and a new way of working crystallised around the SRF: -

> *I can see it becoming embedded, it's just a habit. So you say, "I've sent this off", and then you need somebody to, so what we will do, we've now got somebody in the system who will go, email everybody, "This has been sent", forty eight hours later. "Oh, it's not back", or, "Oh, it's back, here it is", and then I will go, "Oh, let's discuss it", and then she will say, "Oh, that's interesting". You know, so it's a sys… it's a habit as opposed to…. because it's not normal [laughs]. (Site 1)*

Once more, there is a clear dialogue with the NPT mechanism of "Collective action". In some sites a new assemblage of people and processes came together to enable the SRF's insights into transmission to be translated into IPC action.

**Meetings matter and establish a chain of action.** Echoing the theme above, this theme focuses on the ways that, in some sites, communications and connections coalesced and became routinised through regular and timely meetings. The two extracts below show contrasting aspects of this theme. Firstly, an example from a site in which the meetings seemed to work to support the SRF translating to changes to IPC; and a second where key players were not present within these meetings and the interprofessional chain of action across the whole multidisciplinary team (MDT) could not be established and exercised: -

> *You'd say, "This is what we've got", and report back to the entire group, and it becomes part of the agenda, like under typing whole genome sequencing, "This is what our reports have said", and then the team decides what that means. I think that's the only way that it's really going to work, otherwise it's just an interesting thing for those of us who've got a niche interest. (Site 1, 928)*

> *And so that's probably an important thing to note here, is that, yes, you've got an infection control nurse and you've got a virologist and a surveillance officer in this MDT, but where are our clinicians and the managers from the area who are actually the ones in, on the ground, you know, on the shop floor, who are dealing with these outbreaks and who are best placed to probably take this information and take these actions forward? (Site 4, 490, p. 17)*

This theme once more relates strongly to NPT's "Collective action". In the sites in which teams worked to formalise communication and connections, these meetings functioned as a means by which the wide variety of people collectively responsible for implementing the SRF could operate as a cohesive unit.

**Time criticality and the acknowledged value of the SRF.** This minor and somewhat complex theme, relates to the way in which over time – as the SRF was fully understood – staff realised that that there was only a brief window in which its output was probably useful to reduce new nosocomial infections: -

> *I mean, we have discussed it at some of our local meetings, but it's again it's back to the usefulness of it. It's very interesting to sit and look at it and, but if it's not current to what you're doing there's little point in sharing it with anybody else, is there? (Site 4, 299, p. 7–8)*

This theme relates clearly to NPT domains "Coherence" and "Cognitive participation". "Coherence" relates to practitioners understanding of how and why the intervention is being implemented. For the SRF to succeed, there was a reliance on staff understanding both the form itself and the need for a rapid response to it. Thoroughly understanding the SRF, however, also precipitates an acknowledgement of how important turnaround time was for it to be useful. This sense of understanding the SRF and its time-dependent diminishing returns relates to NPT's "Cognitive participation". The theme also resonates with NPT's "Reflexive monitoring"; it ascribed the extent to which the network of people surrounding the SRF were effectively working together to ensure its success.

**Understanding the importance of SRF attribution.** This minor and complex theme captures the way that, for some staff, the effects of the SRF were intangible yet for others they were rendered visible. The SRF didn't particularly change what some IPC staff did as it only changed the location and intensity of what they were doing. In other words, the repertoire of IPC behaviours remained the same. The two contrasting quotes below from the same site show this heterogeneity of experience: -

> *But my view throughout this is that actually we're not doing anything differently for an infection prevention and control, we're looking at standard infection control precautions and transmission-based infection control precautions, they have been there for a very, very long time, it's just the application of them has probably come to the fore. (Site 5)*

> *I think it was an eye-opening study, because we got to clearly see what the strains were, we got the information really quickly, but with other reports, or other methods, or other testing, it just wouldn't have had the same effect for us as a team and our knowledge of what's actually going on with our patients (Site 5)*

Accordingly, for some, differentiation between their usual duties, and duties guided by the SRF, was difficult to grasp. This relates to the NPT construct of "Coherence". Lack of attribution of the SRF to changing IPC behaviours could be considered the result of the 'layers' of people involved in the process by which the SRF led to meaningful action (i.e., the chain of actions) rendering the SRF invisible to some. The overwhelming context of the Alpha variant and the pressures on both the IPC team and the hospital could also have obscured this attribution. In the future ensuring people know that changes to IPC are being driven by the SRF, and that any effective reductions in infections are associated with an SRF will be vital to maintaining its use over time.

## RQ3. How did people using the SRF imagine its effect on trial outcomes?

**Partial success: the imagined outcomes of implementing the SRF.** The final theme within our analysis relates to the ways participants talked about the assumed rather than measured outcomes from the implementation of the SRF during the trial. Perhaps unsurprisingly, given both the facilitative aspects of the context and the reported gravity of implementation challenges, there was a sense of partial success at the time of data collection. Although '*Several bars*' (Site 3) were thought to have been met, '*it's all worked fairly well once we've ironed out the little initial creases*' (Site 2*)* there was also an acknowledgment of its likely limitations: -

> *I think it's got a lot of potential for the future. I think that, you know, as I say this is the first type of study that we've been involved in like this and I think there are lots of lessons to be taken away and learnt from this, and you've got to start somewhere, you know, because nothing's*

*perfect right from the very beginning. So I think it's of value and I think it's worth, I'm glad*
*we've been in it from the beginning. I think it's a valuable experience for our Trust and for our*
*team, but its impact on patient care is not there yet. It needs refining (Site 4, 299, p.8)*

Given the context of a short term non-randomised trial, rather than long-term normalisation of the SRF within usual practice, many key aspects of NPT's outcome constructs were not directly relevant (i.e., 'Intervention Performance', 'Relational Restructuring', 'Normative Restructuring', and 'Sustainment'). However, there is a sense in which the earlier analysis of mechanisms does capture some emergent prototypical implementation outcomes. In those sites in which chains of action formalised through effective communication, new connections and meetings, there is a sense in which relational restructuring was beginning and in which normative restructuring was occurring.

## Discussion

This study aimed to systematically explore the implementation of a simple SRF designed to reduce nosocomial SARS-CoV-2 transmission within the UK hospital setting within an unfolding pandemic. This focus on implementation was vital, as our previous work had highlighted that the SRF was acceptable to diverse staff and that it was likely to work in the way it was intended [11]. Yet our non-randomised prospective study which trialled the relative impact of different target SRF turnaround times (i.e., rapid vs slow) showed mixed results [12] highlighting the need for the systematic investigation of SRF implementation. Here we have presented a detailed analysis of the implementation of the SRF and useful learning for future studies. Given SRFs are novel ways of enhancing infection prevention and control, this study is the first of its kind to explore in depth and detail the challenges and success of SRF implementation.

Using inductive thematic analysis, we have outlined a series of data-driven themes which together recount what our diverse participants recalled about their attempts at implementing the SRF across five differing UK hospitals around the peak of the Alpha variant of SARS-CoV-2. Our themes depicted an overall picture of various teams striving to initiate, use and embed an innovative and promising approach to enhance IPC. Overall, the analysis suggested that, against considerable odds, in some sites, they managed to do so. Given our particular analytic focus on implementation, we then explored our data-driven findings in relation to key theoretical constructs of NPT – an approach developed to explicitly theorise the implementation of interventions and how processes of implementation became normalised over time and within particular contexts. Our work with NPT had three focal points: context, mechanisms and outcomes.

### Key findings

The dialogue between our findings and NPT *contextual* constructs was fruitful – it enabled a grasp of how the context shaped the reported struggles to adapt and accommodate the SRF into routine work. Initially, the alignment of structural and professional capability and opportunity afforded the implementation of the SRF. However, the potential for adaption to embed the SRF diminished dramatically as a peak in COVID-19 numbers presented overwhelming challenges to hospital and IPC function. In this way the specific context of the COVID pandemic both enabled and constrained the implementation of the SRF. The dialogue with NPT's *mechanistic* constructs and our data driven findings highlighted similarities between the implementation of the SRF and many other innovative interventions [25]. Our findings clearly resonated with key implementation mechanisms of "Coherence",

"Collective action" and "Reflexive monitoring". "Cognitive participation" (the degree to which commitment to implementation relates to beliefs in an intervention's potential) however, varied in relation to how fast the SRF could deliver insights into nosocomial infection. Along these lines, implementation worked when people understood the overall endeavour and when they came together to make things happen across the hospital. This sense of things coalescing for a shared purpose was fragile given it demanded people to overextend themselves at a time when hospital function was severely overstretched. Further, if a feedback loop could not be established showing the added value of the SRF, its sustained implementation seemed unlikely. In relation to NPT's *outcome* constructs, there was less conceptual purchase, as could be expected from a narrowly focussed process evaluation of an intervention delivered solely within a trial. Despite this, and even within the timeframes of the trial, a series of useful implementation outcomes began to arise within some sites (e.g., establishing routine meetings, ways of communicating).

## Implications for implementing SRFs

We detailed a series of implications for the future implementation of SRFs (see Table 2) which should be useful for building an evidence base around the use of SRFs within and outside pandemics. Our analysis suggested that it is important to consider particular aspects of the context in which SRFs might be used. Given the magnitude of the impact of the outbreak of the Alpha variant upon healthcare capacity, it is salient to ask whether this was the optimal time for evaluating the impact of SRFs as they might be implemented in the longer-term. Further appraisal of SRFs needs to consider government and stakeholder interest as well as the practicalities of available infrastructure and epidemiological trends that persist beyond the initial stages of the SARS-CoV-2 pandemic. To give one example, where an outbreak is highly localised, it may be possible for resources to be transferred in to assist capacity breaches constraining the embedding of SRFs; however, where capacity is stretched across the entire healthcare system this may be less feasible. Our findings also suggest the need for maintaining a sequencing infrastructure between infection outbreaks and pandemics. Thinking beyond the short-term intensity of effort that the COVID pandemic brought about, there is an ongoing need to identify and support networks of people with an interest in using applied WGS for controlling nosocomial infectious disease.

Table 2 also details a series of implications that stem from what we learned about effective mechanisms across the five sites. These insights may prove valuable for others implementing SRFs in their own settings within and outwith future pandemics. Our analysis points to the need to deliver a package of education and engagement work with diverse hospital staff in order to optimise the use of SRFs in the future. This work should ensure an extensive range of staff within the hospital are aware of SRFs and how they work (particularly issues of time criticality, the importance of rapid turnaround time (e.g., ≤ 2 days) from sampling to IPC action, and the centrality of co-ordinated and collective action to embedding SRFs within the hospital system). Staff should also come to understand their particular roles and responsibilities with regard to the SRF but also appreciate how the SRF affects other diverse teams within the hospital. Work around engaging staff in the SRF must also include a specific focus on the generation of local plans and processes to ensure SRF insights are able to be rapidly translated into IPC action (e.g., effective data sharing and timely routinised meetings across teams and professions). Within meetings and other communications, there should be a clear and consistent focus on monitoring and reducing turnaround time and also a celebration of any SRF-induced success at reducing nosocomial infection. In this way, staff are encouraged to understand that the SRF *can* and *does* work. This is important because it provides positive

reinforcing feedback about the value of diverse teams working together and delivering rapid turnaround times.

## Strengths and limitations

The strengths of the study relate to collecting in-depth qualitative data from a wide range of different professionals who had recently worked with the SRF within the COVID-19 pandemic itself. The timing of interviews ensured that recall was good and the breadth of professionals afforded a rich picture of staff and hospital experience the regarding the challenges and benefits of implementing the SRF. Another strength includes moving beyond a merely descriptive thematic analysis of our data and instead connecting our data-driven themes to NPT which enriched our understanding of SRF implementation. Moreover, unlike many other studies using NPT to explore implementation, a further strength is that we articulated a series of detailed practical implications for future work with SRFs and nosocomial infection.

A limitation of the study was the temporal time frame of data collection, largely taking place within weeks of each site delivering SRFs rapidly and at the peak of the Alpha variant within the COVID-19 pandemic in the UK. This presented unique challenges in collecting data on the way the SRF worked when our findings themselves suggest it took time for the SRF to be understood and embedded. Embedding the rapid delivery of the SRF over a longer period and exploring longer term issues of implementation may give a richer source of insights into how to optimally use SRFs. Another limitation of the study relates to our reliance on interview-derived qualitative data alone to understand implementation. If circumstances had permitted, ethnographic observation or focussed quantitative work could have enhanced our findings. Equally, data was collected within only five of the fourteen sites involved in the larger study. Although these were selected to represent heterogeneous study sites and offered a varied and substantial sample, this may not have captured perspectives across the whole trial. However, the findings were shared and discussed in summer 2021 with a far broader range of staff involved in using the SRF from across all trial sites. Further, the deep burden of COVID-19 on health care professionals across the workforce at the time of data collection should not be overlooked. It may have influenced attitudes towards both the SRF itself, and indeed participation within this study. Finally, further limitations include the pathogen specificity of the study. We anticipate that using an SRF for bacterial or fungal infections would be different. Although the challenge of translating genomic insights into IPC responses would be similar, the time sensitivity of the importance of the turnaround time would be lessened given the typically longer incubation periods and slower pace of transmission events typically associated with either bacteria or fungal infections.

## Conclusion

This is the first paper, globally, to explore the implementation of SRFs to reduce nosocomial infection within the hospital setting. Using rich qualitative data and a theoretical lens, this paper has highlighted how, in some places, it was possible to implement the SRF, make a perceived difference to IPC and potentially reduce nosocomial infection, despite the COVID-19 pandemic. Together, with our other work, we have found that the use of SRFs in hospitals to reduce nosocomial infection is acceptable, feasible and potentially impactful within a five-day turnaround time, even in extreme circumstances across UK hospitals. Critically, however, the use of NPT has informed the importance of considering the context and mechanisms of implementing SRFs, particularly regarding the passion, will, knowledge and understanding of the staff in relation to the SRF, having the hospital infrastructure and capacity to act, and establishing communication and connections across staff to enact a chain of action.

## Acknowledgements

We would like to particularly acknowledge the support of NHS Greater Glasgow and Clyde Clinical Research Facility.

We also acknowledge the support of the independent members of the Joint Trial Steering Committee and Data Monitoring Committee (TSC-DMC): Prof Marion Koopmans (Erasmus MC), Prof Walter Zingg (University of Geneva), Prof Colm Bergin (Trinity College Dublin), Prof Karla Hemming (University of Birmingham), Prof Katherine Fielding (LSHTM). As well as TSC-DMC non-independent members: Prof Nick Lemoine (NIHR CRN), Prof Sharon Peacock (COG-UK). We would also thank members of COG-UK who have directly supported the study: Dr Ewan Harrison (Cambridge University), Dr Katerina Galai (PHE), Dr Francesc Coll (LSHTM), Dr Michael Chapman (HDR-UK), Prof Thomas Connor and team (Cardiff University), Prof Nick Loman and team (University of Birmingham). We also thank the COG-UK Consortium and the UK National Institute for Health Research Clinical Research Network (NIHR CRN). In addition thanks to Thushan de Silva and David Partridge (University of Sheffield), Alison Holmes (Imperial), Laura Shallcross (UCL), Emma Thomson (University of Glasgow), Christine Peters (Greater Glasgow and Clyde), Nicolas Machin (Manchester NHS trust)

## Author contributions

**Conceptualization:** Paul Flowers, Ruth Leiser, Fiona Mapp, Judith Breuer.

**Formal analysis:** Paul Flowers, Ruth Leiser, Julie McLeod.

**Funding acquisition:** Judith Breuer.

**Investigation:** Fiona Mapp, James Blackstone, Gaia Nebbia, James R Price, Luke B Snell, Tranprit Saluja.

**Methodology:** Judith Breuer.

**Supervision:** Paul Flowers.

**Writing – original draft:** Paul Flowers, Ruth Leiser, Julie McLeod.

**Writing – review & editing:** Paul Flowers, Ruth Leiser, Fiona Mapp, Oliver Stirrup, James Blackstone, Christopher JR Illingworth, Gaia Nebbia, James R Price, Luke B Snell, Tranprit Saluja, Judith Breuer.

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
