## [Decision Letter · Decision Letter 0]

18 Sep 2024

PONE-D-24-23160Insights into the implementation of a whole genome sequencing report form (SRF) to reduce nosocomial SARS-CoV-2 within UK hospitals: a qualitative process evaluation using normalisation process theoryPLOS ONE

Dear Dr. flowers,

Thank you for submitting your manuscript to PLOS ONE. After careful consideration, we feel that it has merit but does not fully meet PLOS ONE’s publication criteria as it currently stands. Therefore, we invite you to submit a revised version of the manuscript that addresses the points raised during the review process.

I would like to sincerely apologise for the delay you have incurred with your submission. It has been exceptionally difficult to secure reviewers to evaluate your study. We have now received two completed reviews; the comments are available below. The reviewers have raised significant scientific concerns about the study that need to be addressed in a revision.

Please revise the manuscript to address all the reviewer's comments in a point-by-point response in order to ensure it is meeting the journal's publication criteria. Please note that the revised manuscript will need to undergo further review, we thus cannot at this point anticipate the outcome of the evaluation process.

We look forward to receiving your revised manuscript.

Kind regards,

Miquel Vall-llosera Camps

Senior Staff Editor

PLOS ONE

2. In the ethics statement in the Methods, you have specified that verbal consent was obtained. Please provide additional details regarding how this consent was documented and witnessed, and state whether this was approved by the IRB

“This work was supported by funding from the Medical Research Council (MRC) part of UK Research & Innovation (UKRI), the National Institute of Health Research (NIHR) [grant code: MC_PC_19027], and Genome Research Limited, operating as the Wellcome Sanger Institute.”

6. PLOS requires an ORCID iD for the corresponding author in Editorial Manager on papers submitted after December 6th, 2016. Please ensure that you have an ORCID iD and that it is validated in Editorial Manager. To do this, go to ‘Update my Information’ (in the upper left-hand corner of the main menu), and click on the Fetch/Validate link next to the ORCID field. This will take you to the ORCID site and allow you to create a new iD or authenticate a pre-existing iD in Editorial Manager.

7. Your ethics statement should only appear in the Methods section of your manuscript. If your ethics statement is written in any section besides the Methods, please delete it from any other section.

Reviewers' comments:

Reviewer's Responses to Questions

**Comments to the Author**

1. Is the manuscript technically sound, and do the data support the conclusions?

Reviewer #1: Yes

Reviewer #2: Yes

2. Has the statistical analysis been performed appropriately and rigorously? 

Reviewer #1: N/A

Reviewer #2: N/A

3. Have the authors made all data underlying the findings in their manuscript fully available?

Reviewer #1: No

Reviewer #2: Yes

4. Is the manuscript presented in an intelligible fashion and written in standard English?

Reviewer #1: Yes

Reviewer #2: Yes

5. Review Comments to the Author

Reviewer #1: The article "Insights into the implementation of a whole genome sequencing report form (SRF) to reduce nosocomial SARS-CoV-2 within UK hospitals: a qualitative process evaluation using normalisation process theory" evaluates the implementation and effectiveness of an SRF to reduce hospital-acquired SARS-CoV-2 infections. The study uses normalisation process theory (NPT) to understand how the SRF was adopted in various hospital settings during the COVID-19 pandemic. The findings highlight both enabling factors, such as the passion of healthcare professionals and the existence of effective communication channels, and constraining factors, such as the overwhelming impact of the Alpha variant, which limited infection prevention and control (IPC) capacities. The authors conclude that while the SRF shows potential for enhancing IPC activities, the specific context of the trial and the Alpha variant surge limit the generalizability of the results.

1. The use of in-depth interviews and inductive thematic analysis is appropriate for the process evaluation. However, more detailed information on the interview guide and the coding process would enhance the transparency and reproducibility of the study.

2. While the authors acknowledge the limitations in generalizability due to the trial context and the pandemic's dynamics, it would be beneficial to discuss potential strategies for scaling up the SRF in more stable conditions.

Reviewer #2: Flowers et al. present detailed qualitative data on the implementation of a sequence report form (SRF) for COVID-19 across 14 UK hospitals, with a focused analysis on five for this paper. They contextualize the challenges of implementing the SRF, building on their prior studies that showed mixed results regarding its impact on transmission reduction. Their findings offer practical recommendations for improving the implementation and communication of such data, with high relevance for infection prevention and control (IP&C) departments. Below, I suggest a few clarifications and expansions, particularly regarding the generalizability of genomic epidemiology.

Major Comments:

Contextualizing the use of SRF during a viral pandemic: My primary suggestion is that the authors provide greater context by emphasizing that the SRF was used specifically to detect and stop viral transmission during a pandemic. The application of genomic epidemiology for IPC may differ significantly when applied to other pathogens, such as bacteria or fungi, where the transmission dynamics and staff feedback would likely be very different from those reported for this SRF, especially given the unique circumstances of the Alpha variant stage. For example, the timeline of bacterial outbreaks may be protracted in which turnaround time is not as critical as viral outbreaks in which incubation and transmissibility are shorter in duration. The authors could address this distinction in both the introduction and the limitations sections.

Clarification of SRF's impact Page 16, Line 20: The staff interviewed mentioned “it’s” made a difference. Was the individual specifically referring to the SRF, or could this have been a reference to the enhanced IPC response during the initial waves of COVID-19? For example, it’s possible that the stronger connections formed during the pandemic response, rather than the SRF alone, led to improved outcomes. If the former is the case, perhaps the authors could clarify by adding [SRF] to avoid misinterpretation, as these two interpretations could lead to vastly different understandings of the SRF's impact.

Page 18, Line 10: This staff member’s comment, classified under ‘collective action,’ notes that ‘key players are not present,’ but it reads as if they are suggesting that unless frontline healthcare workers are on board, the SRF cannot be successfully implemented. In other words, even if the process as a whole is effective, unless nurses at the bedside change their practices, the IPC interventions cannot lead to sustained changes. This could be clarified.

Page 19, Line 17: “Understanding the importance of SRF attribution” is crucial, as it directly impacts the perceived effectiveness of interventions, both during the trial and in long-term implementation. If staff believe that the SRF is not driving meaningful change, it’s unlikely that the intervention will be sustained over time. The authors could elaborate on contextual barriers that may have discouraged staff from attributing changes to the SRF.

Minor Comments:

Page 24, Line 12: Just a comment, but this section is critical for IP&C departments to have takeaways on how to better implement an SRF at their own facility.

6. PLOS authors have the option to publish the peer review history of their article (what does this mean? ). If published, this will include your full peer review and any attached files.

**Do you want your identity to be public for this peer review?** For information about this choice, including consent withdrawal, please see our Privacy Policy .

Reviewer #1: No

Reviewer #2: No

---

## [Author Response · Author response to Decision Letter 1]

11 Dec 2024

This has been uploaded as a separate document 'response to reviewers'

---

## [Decision Letter · Decision Letter 1]

10 Mar 2025

Insights into the implementation of a whole genome sequencing report form (SRF) to reduce nosocomial SARS-CoV-2 in UK hospitals within an unfolding pandemic: a qualitative process evaluation using normalisation process theory

PONE-D-24-23160R1

Dear Dr. Flowers,

We’re pleased to inform you that your manuscript has been judged scientifically suitable for publication and will be formally accepted for publication once it meets all outstanding technical requirements.

Kind regards,

Nivedita Jaiswal

Academic Editor

PLOS ONE

Additional Editor Comments (optional):

Reviewers' comments:

Reviewer's Responses to Questions

**Comments to the Author**

1. If the authors have adequately addressed your comments raised in a previous round of review and you feel that this manuscript is now acceptable for publication, you may indicate that here to bypass the “Comments to the Author” section, enter your conflict of interest statement in the “Confidential to Editor” section, and submit your "Accept" recommendation.

Reviewer #1: All comments have been addressed

2. Is the manuscript technically sound, and do the data support the conclusions?

Reviewer #1: Yes

3. Has the statistical analysis been performed appropriately and rigorously? 

Reviewer #1: N/A

4. Have the authors made all data underlying the findings in their manuscript fully available?

Reviewer #1: No

5. Is the manuscript presented in an intelligible fashion and written in standard English?

Reviewer #1: Yes

6. Review Comments to the Author

Reviewer #1: I would like to thank the authors for addressing my comments. While I have no further comments, the manuscript can still be further evaluated by the editorial office.

7. PLOS authors have the option to publish the peer review history of their article (what does this mean? ). If published, this will include your full peer review and any attached files.

**Do you want your identity to be public for this peer review?** For information about this choice, including consent withdrawal, please see our Privacy Policy .

Reviewer #1: No

---

## [Editor Report · Acceptance letter]

PONE-D-24-23160R1

PLOS ONE

Dear Dr. flowers,

I'm pleased to inform you that your manuscript has been deemed suitable for publication in PLOS ONE. Congratulations! Your manuscript is now being handed over to our production team.

Kind regards,

on behalf of

Dr. Nivedita Jaiswal

Academic Editor

PLOS ONE